# Antibacterial Activity of Rainbow Trout Plasma: In Vitro Assays and Proteomic Analysis

**DOI:** 10.3390/ani13223565

**Published:** 2023-11-18

**Authors:** Toita Mizaeva, Kalimat Alieva, Eldar Zulkarneev, Stanislav Kurpe, Kseniya Isakova, Svetlana Matrosova, Ekaterina Borvinskaya, Irina Sukhovskaya

**Affiliations:** 1G. N. Gabrichevsky Research Institute for Epidemiology and Microbiology, 125212 Moscow, Russia; mizaeva@gabrich.ru (T.M.); kallybagandova@mail.ru (K.A.); 2Plague Control Center, Federal Service on Consumers’ Rights Protection and Human Well-Being Surveillance, 119121 Moscow, Russia; zulkarneev_er@apc-rpn.ru; 3Institute of Biochemistry after H.Buniatyan National Academy of Sciences of the Republic of Armenia, Yerevan 0014, Armenia; 4Northern Water Problems Institute of the Karelian Research Centre of the Russian Academy of Sciences, 185000 Petrozavodsk, Republic of Karelia, Russia; isakovakv@krc.karelia.ru; 5Institute of Biology, Ecology and Agricultural Technologies of the Petrozavodsk State University, 185000 Petrozavodsk, Republic of Karelia, Russia; matrosovasv@petrsu.ru; 6Institute of Biology, Irkutsk State University, 664003 Irkutsk, Russia; borvinska@gmail.com; 7Institute of Biology of the Karelian Research Centre of the Russian Academy of Sciences, 185000 Petrozavodsk, Republic of Karelia, Russia

**Keywords:** plasma antimicrobial activity, *Flavobacterium psychrophilum*, *Aeromonas hydrophila*, rainbow trout, LFQ, label-free quantification, proteomics

## Abstract

**Simple Summary:**

The blood plasma of fish can acquire antibacterial activity against a wide range of microorganisms. In our study, when we exposed the pathogenic bacterium *Aeromonas hydrophila* in vitro to blood plasma from naturally infected rainbow trout, we observed agglutination, changes in cell morphology, growth inhibition, and an arrest of protein synthesis in the bacterial cells. Importantly, these effects occurred without causing significant damage to the cell surface. Proteomic analysis revealed that known immune proteins were both up- and down-regulated in the plasma of infected trout compared to healthy fish. A closer examination of the fish proteins that could be retained on the bacterial cells revealed that plasma with high antimicrobial activity contained specific proteins with different biological activities, including antibacterial properties, such as immunoglobulins and ladderlectins. Interestingly, the natural immune defence mechanism in trout plasma, known as the membrane attack complex of the complement system, did not appear to assemble efficiently on the bacteria in the experiment. These findings contribute to our understanding of how plasma proteins defend against pathogenic bacteria.

**Abstract:**

The objective of this study was to investigate the bactericidal activity of blood plasma from cultured rainbow trout obtained from two different fish farms. Plasma from trout naturally infected with the bacterial pathogen *Flavobacterium psychrophilum* was found to inhibit the growth of *Aeromonas hydrophila* in vitro. Incubation of *A. hydrophila* in bacteriostatic trout plasma resulted in agglutination and growth retardation, without causing massive damage to the cell membrane. The proteome of the plasma with high antimicrobial activity revealed an abundance of high-density apolipoproteins, some isoforms of immunoglobulins, complement components C1q and C4, coagulation factors, lectins, periostin, and hemoglobin. Analysis of trout proteins retained on *A. hydrophila* cells revealed the presence of fish immunoglobulins, lectins, and complement components on bacteria whose growth was inhibited, although the native membrane attack complex of immunised trout plasma did not assemble effectively, resulting in a weak bactericidal effect. Furthermore, this study examined the bacterial response to trout plasma and suggested that the protein synthesis pathway was the target of antimicrobial proteins from fish blood. Taken together, these findings illustrate the advantages of the affinity approach for understanding the role of plasma proteins in host defence against pathogens.

## 1. Introduction

The ability of body fluids and secretions to kill bacteria is an important aspect of immune defence. Humoral immunity factors are particularly important in teleosts, as evidenced by the presence of many antimicrobial compounds in their blood, including taxon-specific molecules [1,2,3]. At the same time, the basal levels of antimicrobial factors in fish blood are low; therefore, the innate immunity in fish is usually inducible [4].

Several biochemical mechanisms underlie the bactericidal activity of biological fluids in fishes. A large group of antimicrobial peptides, including lysozymes, defensins, cathelicidins, hepcidins, histone-derived peptides, and fish-specific piscidins, can directly lyse pathogenic cells by penetrating and perforating their cell membranes [2,3,5]. Another sophisticated mechanism for killing bacteria is the assembly of a membrane attack complex (MAC) on the cell surface. Some immune proteins and peptides bind to intracellular targets and inhibit nucleic acid and protein synthesis, disrupt cell wall formation, or induce oxidative stress [6,7,8,9,10]. Histones and immunoglobulins can also stick bacterial cells together, preventing the spread of pathogens within the organism [11,12].

Plasma proteins can exert bacteriostatic effects without direct contact with pathogens by inhibiting the activity of bacterial secretions or reducing the availability of growth factors. For example, protease inhibitors such as alpha-1-antiproteinase and alpha-2-macroglobulin are known to effectively suppress bacterial digestion and dissemination [13,14,15], whereas metal-binding proteins deplete available iron, which is essential for bacterial growth [16,17].

Proteins are a key class of molecules in immunological defence because of their ability to form structures that selectively recognise foreign antigens. Typically, a cascade of immune reactions is triggered via physical contact of host proteins with foreign ligands, signalling the onset of invasion. According to this, many proteins and their native complexes from different animals that are capable of binding to bacterial cells could be promising sources of new immune effector molecules. The functional group of proteins, known as opsonins, which can bind to antigens and non-specific molecular patterns associated with pathogens, are of particular interest because they play a crucial role in initiating the immune response. Of these, immunoglobulins are particularly noteworthy for their wide range of immune functions, including the ability to recognise, tag, and agglutinate bacterial cells, as well as influence bacterial gene expression and neutralise bacterial toxins and essential enzymes [11,18,19,20,21,22,23].

Although many antimicrobial factors have already been discovered in fish plasma, the biochemical and physiological consequences of contact between bacteria and the complex of bacteria-binding proteins present in the blood remain poorly understood [24]. Previously, a simple approach for screening potential immunomodulators was developed using live bacteria as bait for affinity-bound proteins from biological fluids. This approach was applied by Dong et al., 2017 [1], who used *Edwardsiella tarda* to purify potential antimicrobial proteins from the serum of the turbot *Scophthalmus maximus*. Other studies have examined the proteins from the gills and liver of the speckled hind *Epinephelus drummondhayi* that bind to *E. tarda* and *E. piscicida* [25,26], as well as shrimp serum proteins that interact with *Vibrio parahaemolyticus* and human plasma factors that bind to *Streptococcus pyogenes* [27,28]. The high-throughput mass spectrometry used in these studies provides unique information on protein–protein interactions between microorganisms and the host as well as real-time monitoring of adaptive metabolic rearrangements of pathogens during invasion. This approach is therefore very promising for screening previously uncharacterised proteins with antimicrobial activity in biological fluids from different species.

In this study, we investigated the proteins involved in the natural antibacterial activity of blood plasma from cultured rainbow trout, *Oncorhynchus mykiss*. First, we studied the growth dynamics, morphology, and protein profile of the fish pathogen *Aeromonas hydrophila* when incubated in plasma from trout from two different fish farms. Second, a comparative analysis of fish plasma samples with different levels of antimicrobial activity was carried out. To identify molecules with signalling functions and bactericidal activity, the study design also included an investigation of trout proteins interacting with live *A. hydrophila* in vitro. Finally, changes in the pathogen proteome in response to contact with the host environment were studied to elucidate the mechanisms of action of fish antimicrobial factors.

## 2. Materials and Methods

### 2.1. Blood Plasma Collection

Blood plasma was collected from two independent populations of diploid rainbow trout obtained from fish farms between August and September 2021. All fish were sampled from one cage on the same day at each farm using a fish net. Fish (*n* = 4) from the first farm (site N, Republic of Karelia, Northwest Russia; N: 61.5°, E: 33.6°) weighed 1–1.5 kg and were characterised as relatively healthy, with no external signs of ichthyopathology. Fish (*n* = 10) from the second fish farm (site A, Eastern Siberia, Russia, N: 52.3°, E: 104.3°) weighed 100–450 g and showed signs of chronic bacterial disease of an unknown aetiology. The sick animals on this farm were characterised by intestinal inflammation, rectal prolapse, and skin hyperemia (Figure 1).

On the day of sampling, the fish were anaesthetised with clove oil emulsified in water (0.2 mL/L, 5 min). Blood was collected from the caudal vein using BD Vacutainer^®^ Plus vacuum tubes containing EDTA (Heidelberg, Germany). The blood was then centrifuged at 1610× *g* for 3 min and the plasma was collected. Individual samples were pooled, aliquoted, and stored at −20 °C until analysis.

### 2.2. Molecular Identification of the Rainbow Trout Pathogen

Total DNA of diseased fish from farm A was isolated from anal sphincter tissues using a DiaGene DNA extraction kit (Dia-M, Moscow, Russia). For this, tissue samples were taken from trout in the same cage with a swollen anus (*n* = 5) or from relatively healthy fish (*n* = 4) with no external signs of pathology. The sphincter was dissected under sterile conditions with scissors and a piece of tissue was frozen before analysis at −80 °C. Total DNA was isolated from 50 mg of homogenised tissue in the lysis buffer using the FastDNA SPIN Kit (MP) according to the manufacturer’s instructions.

A mixture of eight primers was used for amplification of the 428 bp hypervariable V3–V4 region of the 16S ribosomal RNA gene (Appendix A). Amplification was performed in a volume of 25 μL of a mixture containing 5 μL KTN-mix (Evrogen, Moscow, Russia), 2 μL primer mixture, and 0.5 μL SYBR (Evrogen) in a CFX96 Touch real-time amplifier (Bio-Rad, Hercules, CA, USA) under the following conditions: initial denaturation for 3 min at 95 °C; 35 cycles of denaturation for 30 s at 95 °C, annealing for 30 s at 57 °C, elongation for 30 s at 72 °C, and a final elongation for 5 min at 72 °C. Amplicons were purified after amplification using AMPure XP magnetic particles (KAPA Biosystems, Woburn, MA, USA). The resulting pool of libraries was sequenced on the Illumina MiSeq by paired-end reading with the generation of at least 10,000 paired-end reads per sample using the MiSeq Reagent Kit v2 nano and MiSeq v2 Reagent Kit. The PhiX phage library was used to control the sequencing parameters.

The sequencing data obtained were processed using the QIIME 1.9.1 algorithm, which included combining forward and reverse reads, removing technical sequences, filtering sequences with low read reliability scores for individual nucleotides (quality less than Q30), filtering chimeric sequences, aligning reads to the reference 16S rRNA sequence, and assigning sequences to taxonomic units using the Silva database version 132 and Unite v8. An algorithm for the classification of operational taxonomic units (OTUs) with an open-reference OTU (97% classification threshold) was used. Species categorisation was performed using the BLAST algorithm. 

### 2.3. Cultivation and Characterisation of A. hydrophila

*Aeromonas hydrophila* (GKPM-Obolensk strain B-7964) was isolated in autumn 2014 in the Republic of Karelia (Northwest Russia) from the water at a trout farm where an outbreak of aeromonosis was reported. The strain is characterised as motile Gram-negative straight sticks with rounded ends. The temperature range of the strain growth was 10–42 °C, with an optimum at 28 °C. The species identity of the culture was confirmed using MALDI-TOF. Prior to the experiment, the culture was stored in lyophilised form in the collection of the G. N. Gabrichevsky Moscow Research Institute of Epidemiology and Microbiology, Federal Supervision Service for Consumer Rights Protection and People’s Welfare (Moscow, Russian Federation).

A daily culture was obtained as follows: collection strain of *A. hydrophila* was transplanted by looping onto a separate Petri dish with 1.5% Mueller–Hinton agar (MHA; HiMedia, Mumbai, India) and incubated for 18–20 h at 30 °C. Five homogeneous, smooth, convex colonies (1–3 mm) were transplanted by looping into tubes containing 4.5 mL of Brain Heart Infusion Broth (BHI BROTH; Conda, Spain) and incubated for 18 h at 30 °C. Before analysis, 100 μL of the bacterial overnight culture grown in a liquid nutrient medium was added to 2 mL of BHI Broth and germinated to the exponential growth stage with an optical density of a solution of 0.6 units at a wavelength of 600 nm (OD_600_).

### 2.4. Analysis of Antimicrobial Activity in Trout Plasma

On the day of analysis, native trout blood plasma was thawed at room temperature for 20–30 min, resuspended by shaking on a vortex (Heildolph, ReaxTop), filtered using a 0.22 μm syringe nozzle (PES, Merck, Germany) under sterile conditions, and stored on ice. Blood plasma from fish caught at farm A was further designated as AP and plasma from fish caught at farm N was designated as NP. To obtain the low molecular weight fraction of AP and NP plasma with a molecular weight of less than 10 kDa (APf and NPf, respectively), an aliquot of native plasma was filtered using Amicon Ultra-0.5 Millipore concentrators (Sigma-Aldrich, Billerica, MA, USA) on an Eppendorf 5427R centrifuge (Germany) for 10 min at 10,000× *g* and 4 °C. Native AP plasma was denatured by heating in a Gnome solid thermostat (DNA-Technology, Moscow, Russia) for 30 min at 56 °C to monitor the antimicrobial activity of the proteins. The resulting inactivated fish plasma was further denoted as AP°t plasma.

An overnight bacterial culture of *A. hydrophila* was seeded onto a 1.5% MHA agar medium (HiMedia, India). The Petri dish was divided into six sectors, where each sector was filled with 30 µL of native or filtered plasma diluted 1, 10, and 100 times. The Petri dishes were dried for 10–15 min at room temperature and then incubated in a thermostat for 24 h at 30 ± 1 °C. Antimicrobial activity was assessed based on the presence of a growth retardation zone on agar at the plasma application sites.

Trout plasma showing antimicrobial activity was examined for agglutinating activity. A drop of trout plasma and the physiological solution were applied to the surface of a degreased slide. *A. hydrophyla* was added to both drops, removed from the surface of the dense nutrient medium, and dispersed until a homogeneous suspension was obtained. The samples were then gently rocked for 30–60 s at room temperature. The formation of flakes indicates a positive reaction.

### 2.5. Assessing the Growth Dynamics of A. hydrophila

The antimicrobial susceptibility of bacteria to trout plasma collected from different fish farms was assessed in a liquid medium via the turbidimetric method using biomass optical density curves at 620 nm. The activities of the different fractions and dilutions of trout plasma were assessed in a sterile 96-Well Cell Culture Plate (Eppendorf). The wells were filled with 90 µL of different incubation media: AP, AP°t, APf, NP, NPf, Mueller–Hinton broth (MHB), or phosphate buffer, pH 7.2–7.4 (PBS). Then, 10 µL of the prepared bacterial suspension was added to the wells (final concentration of bacteria was approximately 2–4 × 10^5^ CFU/mL). A sterile negative control from the MHB culture medium was also added to the individual wells of the plate without the bacterial culture (uninoculated). The plate was covered with a lid and incubated for 225 min by shaking at 20 °C on an Infinite F50 microplate reader (Tecan, Grodig, Austria). Biomass gain was evaluated by measuring the optical density of the medium at 620 nm every 15 min. Each measurement was performed in triplicates.

After incubation, the replicates of cell suspensions treated with AP, AP°t, or NP plasma were pooled and washed away from unbound trout proteins. For this, bacteria were centrifuged in microtubes for 2 min at 10,000× *g* (centrifuge CM-50 ELMI, Riga, Latvia), the supernatant was removed, and the precipitate was resuspended in 1 mL PBS (pH = 7.4). The sedimentation–suspension series was repeated twice. After the final centrifugation, 1–2 mL of 0.1% SDS per 100 mg of bacteria was added to the precipitate to destroy cells, frozen, and stored until proteome analysis at −70 °C.

### 2.6. Assessing the Viability of A. hydrophila

The effect of trout plasma on *A. hydrophila* viability was assessed by adding 10 µL of the bacteria suspension grown overnight (final concentration 2–4 × 10^5^ CFU/mL) to 90 µL of the medium with different compositions (AP, APf, AP°t, NP, NPf, MHB, or PBS) to the plate wells (three replicates). After 4 h of incubation at 20 °C, the bacterial suspension was collected from the wells, diluted 100 fold with PBS, and 100 µL of the resulting solution was rubbed with a sterile spatula onto the surface of solid 1.5% MHA. The Petri dishes were dried at room temperature for 15 min and incubated for 24 h in a thermostat at 30 ± 1 °C. The cups were removed after 24 h and the number of colony growth units (CFU) was counted.

### 2.7. Assessment of Morphology and Cell Integrity of A. hydrophila

The number of live cells and morphology of *A. hydrophila* after incubation with fish plasma were assessed via fluorescence microscopy after staining the cells with LIVE/DEAD^®^ BacLight Bacterial Viability Kits (L7012, Invitrogen, Eugene, OR, USA). According to the instructions, 90 µL of the cell suspension in the exponential growth stage was precipitated via centrifugation for 7 min at 7000× *g* (CM−50 ELMI centrifuge, Latvia), the supernatant was removed, and the precipitate was resuspended in 10 µL of 0.9% NaCl. Then, 10 µL of the bacterial suspension was added to a test tube containing 90 µL of native bacteriostatic plasma (AP) or PBS (control group) and incubated for 4 h at 20 °C in a Shaker ST−3 Sky line (ELMI, Riga, Latvia). The obtained cell suspensions were diluted 100 times and incubated with propidium iodide and SYTO 9 dyes from the kit at 28°C for 15 min in a thermoshaker. The excess fluorescent dye was then washed with a single precipitation–resuspension in 20 µL of 0.9% NaCl. Aliquots of stained bacteria (10 µL) of each dilution were introduced into a Goryaev chamber (MiniMed, Bryansk, Russia) and the samples were examined at a 20× and 40× magnification using a ZEISS Axio Scope A1 fluorescence microscope (Carl Zeiss Microscopy GmbH, München, Germany). The number of live (green staining) and dead (red staining) cells were counted in five horizontal and 15 diagonal large squares (Appendix A).

### 2.8. Proteomic Analysis of Trout Plasma Proteins

Proteomic research included the following tasks: (1) to compare the blood plasma proteome with high (AP) and low (NP) bactericidal activity from individual fish from farm A (*n* = 4) and N (*n* = 4), respectively (Appendix A); (2) to describe the profile of blood proteins (pool of three experimental replicates) with a potential immune function that adhered to bacterial cells after incubation in AP, AP°t, and NP (hereafter referred to as AP + Bac, AP°t + Bac, and NP + Bac, respectively); (3) and to compare the proteome composition of *A. hydrophila* (pool of three experimental replicates) unexposed (archive B−7964 strain) or exposed to fish plasma (Appendix A).

The native plasma samples were analysed as is, while the bacterial lysates were pretreated for protein isolation. For this, the collected bacterial suspensions were thawed and lysed using a Bandelin Sonopuls HD 2070 ultrasonic probe (Germany) in cold water three times for 30 s (30% power) in a buffer containing 2.5% SDS in 25 mM triethylammonium bicarbonate (TEAB). The lysate was centrifuged for 10 min at 13,000× *g* Eppendorf Centrifuge 5415R (Hamburg, Germany) to remove the foam. Isolated proteins were concentrated on a Microcon Ultracel PL—10 filter to a volume of 100 µL.

Proteome analysis was performed using the equipment of the Human Proteome Basic Facility of the Institute of Biomedical Chemistry (Moscow, Russia). Protein in the samples was measured via the BSA method, for which 1 mL of a reagent containing 1% sodium salt of bicinchoninic acid, 2% Na_2_CO_3_, 0.16% sodium tartrate, 0.4% NaOH, 0.95% NaHCO_3_ (pH = 11.25), and 20 µL 4% CuSO_4_ were added to 30 µL of the sample. The solutions were then stirred and incubated at 56 °C for 20 min on a Thermomixer comfort shaker (Eppendorf, Germany). The soluble protein concentration was determined at 562 nm on a NanoDrop ND−1000 spectrophotometer (Thermo Fisher Scientific, Waltham, MA, USA) using a calibration curve with standard BSA solutions, at concentrations of 0.5, 1, 2, 5, 10, 25, and 50 µg/μL (three replicates).

Tryptic protein cleavage was performed according to the sample preparation protocol on the S trap filter (Suspension Trapping; doi:10.1002/pmic.201300553). The S-trap filter was composed of a quartz insert and reverse-phase C8 sorbent. The sample was acidified, then methanol was added and the resulting suspension was retained in a quartz insert. The sample was washed and hydrolysed and the resulting peptides were eluted and further purified on the reverse-phase sorbent. The peptide eluate (three aliquots) from the collection vial was transferred to a glass vial for subsequent drying in a rotary evaporator at 45 °C. After complete drying, the samples were dissolved in 20 μL of 0.1% formic acid and used for further analysis.

Proteomic analysis of the peptides was performed using an Ultimate 3000 RSLCnano chromatographic HPLC system (Thermo Scientific, Waltham, MA, USA) coupled to a Q-exactive HFX mass spectrometer (Thermo Scientific, Waltham, MA, USA) in three technical replicates for each sample. One microliter of the peptide mixture was loaded onto an Acclaim µ-Precolumn enrichment column (0.5 mm × 3 mm, particle size 5 μm, Thermo Scientific, Waltham, MA, USA) at a flow rate of 10 μL/min for 4 min in isography mode using buffer “C” as the mobile phase (2% acetonitrile and 0.1% formic acid in deionised water). Next, the peptides were separated on an Acclaim Pepmap^®^ C18 HPLC column (75 µm × 150 mm, 2 µm particle size) (Thermo Scientific, Waltham, MA, USA) in gradient elution mode. A gradient was formed with mobile phase A (0.1% formic acid) and mobile phase B (80% acetonitrile and 0.1% aqueous formic acid solution) at a flow rate of 0.3 μL/min. The column was washed with 2% mobile phase B for 10 min, after which the concentration of mobile phase B was linearly increased to 35% in 68 min, then the concentration of phase B was linearly increased to 99% in 2 min; after a 2 min wash at 99% buffer B, the concentration of this buffer was linearly decreased to the initial 2% in 3 min. The total duration of the analysis was 90 min.

Mass spectrometric analysis was performed on a Q-Exactive HFX mass spectrometer in the positive ionisation mode using a NESI source (Thermo Scientific, Waltham, MA, USA). The emitter voltage and capillary temperature were set to 2.1 kV and 240 °C, respectively. Panoramic scans were performed over a mass range of 300–1500 *m*/*z* at a resolution of 120,000. The resolution in tandem scanning was set to 15,000 in the mass range from 100 *m*/*z* to the upper limit, which was determined automatically based on the precursor mass but not more than 2000 *m*/*z*. Precursor ions were isolated within a window of ±1 Da. The maximum number of ions allowed to be isolated in the MS2 mode was set as 40 or less, with the cutoff limit for precursor selection for tandem analysis set as 50,000 units and the normalised collision energy equal to 29. Only ions from z = 2+ to z = 6+ in the charge state were considered for the tandem scan. The maximum accumulation time for the precursor ions was 50 ms, and for the fragment ions, 110 ms. The AGC values for the precursors and fragment ions were set 1 × 10^6^ and 2 × 10^5^, respectively. All measured precursors were dynamically excluded from tandem MS/MS analysis for 90 s.

### 2.9. Bioinformatics Analysis

Protein identification in the mass spectra was performed using MaxQuant v. 1.6.3.4 via the Andromeda search algorithm. UniProt sequence databases UP000193380 (*Oncorhynchus mykiss* proteome) and UP000000756 (Aeromonas hydrophila subsp. hydrophila proteome) were used to identify proteins in the samples. The following search parameters were specified: trypsin cleavage enzyme, accuracy of monoisotopic peptide masses ± 4.5 ppm, accuracy of masses in MS/MS spectra ± 20 ppm, and possibility of missing two trypsin cleavage sites. Methionine oxidation, protein N-terminal acetylation, and cysteine carbamidomethylation were considered as possible and obligatory peptide modifications, respectively. A false discovery rate (FDR) of 1.0% or less was used for reliable peptide spectrum matches (PSM) obtained from MS/MS searches against the protein sequence database. Protein was reliably identified if at least two unique peptides were detected. The analysis of protein abundance was based on label-free quantification (LFQ). The mass spectrometry results are available in ProteomeXchange under the identifier PXD037789, PXD043655, and PXD044468.

Data analysis of the mass spectrometric results was performed using the R programming language. Proteins belonging to the rainbow trout were selected from lists of label-free quantitation (LFQ) proteins and the following proteins were excluded: contaminant proteins, proteins only identified by site, proteins with one unique peptide, and proteins with a score of 10 or less. Proteins with LFQ values less than two of the three technical repeats were filtered out. The LOG2 transformed data were normalised using the quantile normalisation method (R package Limma). Data imputation was performed using the QRILC (quantile regression) method (R package imputeLCMD). 

The differential expression (DE) between the AP and NP plasma samples was analysed using multiple linear models (moderated t-statistics) and empirical Bayes to estimate the prior probability distribution (R package Limma). Proteins with statistically significantly altered expressions were filtered using absolute logFC values greater than 1 and adjusted *p*-values corrected using the Benjamini and Hochberg method of less than 0.05. Qualitative analyses were performed on the pooled samples of bacteria treated with plasma.

Analysis of protein–protein interactions and the enrichment of metabolic pathways and processes was performed using the StringDB web server with clustering [29]. Functional annotation of the proteins of interest from the Gene Ontology and InterPro databases was performed using the BLAST2GO program (BioBam Bioinformatics SL, Valencia, Spain). Clustering of protein isoforms in the proteomes of *O. mykiss* and *A. hydrophila* was performed using the CD-HIT server, with a >70% identity cut for similar proteins [30]. Membrane domains, cleavage regions, signal peptides, and protein localisation were predicted using DeepLoc 2.0, SecretomeP-2.0, DeepTMHMM, and SignalP 6.0 web services [31,32,33,34]. The resulting protein–protein plots were obtained using the StringDB online tools, whereas other plots were generated using the ggplot2 and VennDiagram R packages. 

A search for potentially antigenic *A. hydrophila* sequences from *in vitro* experiments was performed using homology analysis with known virulence factors of *F. psychrophilum*, which is thought to be the causative agent of trout infection in farm A. The amino acid sequence database for *Aeromonas hydrophila* subsp. hydrophila (strain ATCC 7966/DSM 30187/BCRC 13018/CCUG 14551/JCM 1027/KCTC 2358/NCIMB 9240/NCTC 8049; taxonomy ID: 380703, proteome ID: UP000000756) and *Flavobacterium psychrophilum* (strain ATCC 49511/DSM 21280/CIP 103535/JIP02/86, UP000006394, taxonomy ID: 96345) as well as the virulence factor database [35] were used to search for the closest match among the proteins using BLASTp (Blast2GO). The antigenicity of the *A. hydrophila* protein was calculated using the Vaxijen V2.0 tool [36], with the prediction accuracy parameter threshold set at 0.4. The virulence factors of *A. hydrophila* with the highest similarity to those of *F. psychrophilum* that were predicted to be highly immunogenic were selected as potential antigens (Appendix A).

## 3. Results

### 3.1. Antimicrobial Activity of Trout Plasma

A study of antimicrobial activity showed the inhibition of *A. hydrophila* growth by native undiluted blood plasma obtained from trout from farm A suffering from inflammatory bowel disease (antimicrobial plasma, AP; Figure 2). No antimicrobial activity against *A. hydrophila* was observed in the native plasma of healthy trout caught at farm N (non-antimicrobial plasma, NP). The same low level of bactericidal activity was observed in the 10-kDa cut-off filtered plasma (APf and NPf) of fish from both populations studied (Figure 2), suggesting that the inhibitory effect of AP is entirely associated with the protein fraction.

After four hours of incubation in AP, but not NP plasma, *A. hydrophila* lost its ability to divide on the culture media (Figure 2C), although, according to lifetime staining, the membranes of approximately two-thirds of the cells were not destroyed (Figure 3, Appendix A). Thus, factors of AP plasma were shown to be weakly bactericidal and strongly bacteriostatic.

Microscopy revealed significant changes in the morphology of the bacterial cells treated with AP plasma. While *A. hydrophila* grown in the MHB culture medium appeared as individual or fissile 1–3 µm bacilli (Figure 3A,B), bacteria incubated in native AP were of irregular length (shortened or elongated rods) and assembled into chains of at least five cells (Figure 3C,D).

### 3.2. Identification of the Causative Agent of Trout Infectious Disease

Cultured rainbow trout from fish farm A suffered from chronic enteritis and rectal prolapse of an unknown aetiology (Figure 1). Sequencing of the 16S rRNA gene in swollen tissues of the anal sphincter yielded between 2710 and 7833 reads (average 5301), ranging from 83 to 187 individual taxonomic units (OTUs). Profiling of the microbial community in the diseased fish revealed an overwhelming number of reads belonging to the genus Flavobacterium (29–91% of the total, Appendix A). According to BLAST analysis, the closest homologue was *F. psychrophilum*, which was the causative agent of “cold-water disease” in salmonids. In the control group of healthy fish from the same cage, less than 1% of the reads were attributed/assigned to this genus. Meanwhile, members of the taxon Aeromonadales were identified in only half of all fish from farm A and in a maximum of 4% of the total reads (Appendix A). This result suggests that the observed in vitro cross reactivity of AP proteins with *A. hydrophila* is the result of the in vivo immunisation of trout from farm A against Gram-negative *F. psychrophilum*.

Comparative protein sequence alignment of *F. psychrophilum* and known *Aeromonas* sp. virulence factors [37] revealed 45 orthologous proteins with conserved domains (Appendix A). The most similar *F. psychrophilum* and *A. hydrophila* virulence factors with antigenic sequences were mostly membrane proteins that corresponded to structures such as polar flagella or Tap, MSH, and Flp type IV pili, and to functional categories such as motility and adhesion. This suggests that putative *A. hydrophila* antigens in the experiment are surface-localised, and thus, readily available for host immune factors.

### 3.3. Comparative Analysis of the Trout Plasma Proteomes

Blood plasma protein profiles were analysed in trouts with high and low plasma antimicrobial activity collected from two different fish farms (Figure 4). In the screening study, a total of 277 proteins were reliably detected in the trout plasma samples. The majority of the proteins (77%) were present in the plasma of both populations, where peptides from 34 proteins (12%) were identified only in AP plasma with high bacteriostatic activity and 30 (11%) only in NP plasma with low antimicrobial activity (Figure 4A).

Differential expression (DE) analysis was performed to identify proteins whose relative abundance (LFQ) differed between the AP and NP plasma from the sampled individuals (Figure 4B). Of these, 13 proteins were found to be more abundant in the AP proteome and 26 proteins were statistically more abundant in the NP proteome (Figure 4B, Appendix A). The top altered functions and processes in AP, revealed by the comparative GO term enrichment analysis of DE proteins, were related to humoral immune response, including opsonin and complement binding (GO:0006959, GO:0001846, and GO:000184), lipid transport (GO:0019433; GO:0030317; GO:0033344; GO:0034381,GO:0050750; and GO:0120020), gametogenesis (GO:0030317), endopeptidase inhibitor activity (GO:0004866), and the regulation of blood coagulation (GO:0008201; Appendix A).

A notable feature of AP samples was the modification of lipid transport, specifically the up-regulation of apolipoproteins A-IV (A0A060XNR6, A0A060XUR3, and A0A060YKT0) and apolipoprotein M (A0A060YST0) (Figure 4C), with the down-regulation of apolipoprotein B-100 isoforms (A0A060WEF7, A0A060Z757, A0A060W3H4, and A0A060Z709), phospholipid transfer protein (A0A060WYY3), and lipocalin-type prostaglandin D synthase (A0A060YC37). These plasma proteins are involved in the assembly and remodelling of plasma lipoprotein particles, which are responsible for transporting fat through the blood. Since the fish from farm A (up to 500 g weight) were juveniles and the fish from farm N (up to 1.5 kg weight) had already developed gonads, the observed changes in plasma lipoprotein abundance may reflect both dietary and maturity differences between the fish from the two populations studied. The up-regulation of the zona pellucidasperm–binding protein (A0A060XQ20) in NP confirms that the increased lipid transport of fish from farm N may be associated with gametogenesis in mature fish. However, it should be noted that lipoproteins B-100 are multifunctional molecules involved not only in biosynthesis, but also in haemostasis and the regulation of the innate immune response (Reactome MAP-109582, MAP-168898), which were significantly affected in AP compared to NP (Figure 4C, Appendix A). Therefore, these changes may reflect the different immune status of infected and uninfected fish.

A significant difference in the regulation of peptidase activity was also noted between AP and NP plasma, resulting in the enrichment of serine proteases, trypsin domain, and endopeptidase inhibitor activity (STRING CL:45523) among the differentially expressed proteins (Appendix A). Various isoforms of proteinase inhibitors involved in the regulation of coagulation and inflammation, such as kininogen 1 (A0A060WI78), alpha-2-macroglobulin (A0A060WW01), inter-alpha-trypsin inhibitor heavy chain 3 (A0A060YKA6), complement C1r-A subcomponent (A0A061A8T0), and serpin domain-containing proteins alpha-2-antiplasmin (A0A060XQ45) and alpha-1-antitrypsin (A0A060XWS9), were less abundant in plasma with high antimicrobial activity, whereas the endopeptidase inhibitor protein AMBP and complement C4-like protein were found to be up-regulated in AP.

In general, the studied trout from farm A were characterised by an altered expression of haemostatic and fibrinolytic proteins (Figure 4C), with isoforms of fibrinogen (A0A060XZH2, A0A060YWY3), fibronectin (A0A060YV80), alpha-2-antiplasmin (A0A060XQ45), kininogen (A0A060WI78), and plasminogen (A0A060W7T7) down-regulated and alpha-1-microglobulin/bikunin precursor AMBP (A0A060YC43), haemagglutinating lectin (A0A060W1N4), and coagulation factor XIII B (A0A060YM30) up-regulated in AP. Among the genes down-regulated in AP, there was enrichment for terms indicating protein localisation in the fibrinogen complex (GO:0005577, STRING CL:45796) and in the degranulation of platelets (Reactome, MAP-114608; Appendix A). This, together with the elevated levels of plasma haemoglobin (A0A060Z7S3) in the AP, suggests a state of haemolytic and haemorrhagic anaemia in the fish from farm A caused by *F. psychrophilum* infection (Figure 4C).

However, the most significant shifts in expression in the immune protein pool have been reported for apolipoproteins, complement components, lectins, and lytic enzymes (Appendix A). In bacteriostatic AP, the levels of the immunoglobulin variable heavy-chain protein (IGHV, A0A060Y462), uncharacterised C-type lectin (A0A060W1N4), complement component C4 (A0A060YQU6), and coagulation factor XIII B (A0A060YM30, complement factor H family) were significantly higher than those in NP, suggesting the activation of the complement system. Consistent with this, the levels of the anti-inflammatory protein alpha-1-antitrypsin (A0A060XWS9) and lipocalin-type prostaglandin D synthase (A0A060YC37) were reduced in AP. The most significant changes were found for the complement C1q component, the acute phase protein precerebellin (Q9PT14) [38], whose relative level was approximately four times higher in AP than in NP plasma. However, the abundance of another isoform of the cerebellin precursor, A0A060WRX4, did not differ in the plasma of fish from the two populations, suggesting that cerebellin function was not compromised in plasma with low antimicrobial activity.

In contrast to the activation of some inflammatory proteins, the levels of other complement pathway activators, such as C1r-A (A0A061A8T0) and complement factor H-like (A0A060Y1K2), as well as some proteins recognising pathogen-associated molecular patterns (PAMP), such as the C-reactive protein (A0A060YNT5, A0A060XVX4) and intelectin (A0A060ZE00), were lower in plasma with higher antimicrobial activity than NP. Overall, proteins involved in the activation of complement C3 and C5 (STRING CL:46089, CL:45531) were down-regulated in AP compared with NP, suggesting that the humoral immune response in fish from farm A was impaired during chronic *F. psychrophilum* infection (Figure 4C, Appendix A).

### 3.4. Trout Plasma Proteins Interacting with A. hydrophila Cells

Most of the proteins (88%) previously detected in native fish plasma were also found in varying amounts in the washed lysates of *A. hydrophila* incubated in AP or NP (AP+Bac and NP+Bac, respectively; Figure 5A, Appendix A). In addition, 64 fish proteins were identified only in the samples of bacteria treated with trout plasma. These proteins, which bind selectively to bacterial cells, are likely to be minor components of the plasma and could not be detected using intact plasma mass spectrometry due to the technical limitations of the method.

To distinguish the native antimicrobial factors of AP, the bacteria were additionally treated with inactivated AP plasma (AP°t + Bac), which had no effect on bacterial growth. Heating resulted in increased non-specific binding of plasma proteins to bacterial cells (Appendix A); approximately 27% of the identified native plasma proteins adhered to bacteria only after denaturation. Meanwhile, the vast majority of proteins detected in *A. hydrophila* lysates after treatment with native AP and NP were able to bind to bacteria in a denatured form, despite the loss of biological activity (Appendix A). Due to the high level of non-selective protein adhesion, the samples treated with heated plasma were considered uninformative and excluded from further analysis.

Qualitative analysis of the proteome revealed six proteins that were only detected in the lysate of bacteria treated with native AP (as well as in inactivated AP°t), but were absent in bacterial samples incubated in NP (Figure 6). These proteins, in their native conformation, may be candidates for a specific plasma activity against *A. hydrophila*: heavy variable chain of unspecified immunoglobulin (A0A060YVB8), Ig light chain kappa (A0A060Z062), ladderlectin-like protein (A0A060WRB3), complement component C8 gamma chain (A0A060XV51, lipocalin family), properdin-like protein (A0A060XDG4), and actin isoform with unknown function (A0A060W0R3). For all these proteins, except actin, participation in the immune process has been shown. Of these, antigen recognition has previously been described for immunoglobulins and ladderlectins, whereas complement component C8 and properdin are most likely involved in the formation of the membrane attack complex (MAC).

### 3.5. Proteome of A. hydrophila Inhibited by the Antimicrobial Components of AP Plasma

To identify the target proteins for antimicrobial factors in fish plasma, we compared the protein profiles of bacteria treated with AP or NP plasma or grown on the MHB culture medium (hereafter Bac + AP, Bac + NP and Bac + MH, respectively). A total of 771 protein sequences were identified in the proteome of *A. hydrophila* in the experiment (Figure 6A). Of these, only six unique proteins were identified in the proteome of bacteria exposed to bacteriostatic fish plasma: the acyltransferase family protein (A0KNH8), alcohol dehydrogenase (A0KNJ0), iron-siderophore ABC transporter (A0KJP4), adhesin-like protein (A0KIU4), macrolide transporter (A0KMJ3), and peptidase M13 (A0KHJ6). Importantly, according to protein–protein interaction analysis using the StringDB web server, these proteins are not part of a common network of physical interactions, metabolic pathways, or cell compartments. This modest and chaotic rearrangement of protein synthesis suggests that AP plasma does not trigger any specific defence mechanisms in *A. hydrophila* against the host immunity. Furthermore, these proteins were not recognised as *A. hydrophila* virulence factors (Appendix A), confirming the non-specificity of the antimicrobial humoral factors of fish from farm A against bacterial invaders.

Significantly more proteins were not identified in *A. hydrophila* treated with antimicrobial AP plasma, although they were present in the proteomes of bacteria incubated in MHB and NP, which did not affect bacterial growth. This suggests that their synthesis is inhibited by antibacterial factors present in fish plasma. Generally, among the missing proteins, it is possible to highlight a functional cluster consisting of proteins associated with ribosomes and protein biosynthesis (Figure 6B). Along with GTPase Der (A0KJ48) and the ATP-dependent RNA helicase RhlE (A0KKS5), which assists ribosome assembly in bacteria [39,40], the translation initiation factor IF-3 (A0KKP8) binds to the 30S ribosomal subunit and initiates protein synthesis and the 23S rRNA (guanosine 2′-O-) methyltransferase RlmB (A0KG60) modifies 23S rRNA [41,42], but the ribosomal subunit proteins S17, L9, L20, and L23 (A0KF30, A0KG69, A0KKP6, and A0KF23, respectively) were missing from the proteome of AP-treated bacteria. This indicates critical damage to the core of the biosynthetic process in *A. hydrophila* in AP plasma, which is consistent with the observed complete inhibition of bacterial cell growth while maintaining cell membrane integrity (Figure 2 and Figure 3). Notably, enzymes involved in threonine synthesis, such as homoserine dehydrogenase (A0KMM1) and threonine synthase (A0KML9), were deficient in the AP-treated bacteria [43,44]. Notably, enzymes involved in threonine synthesis, such as homoserine dehydrogenase (A0KMM1) and threonine synthase (A0KML9) were in deficiency in AP-treated bacteria [43]. Given the role of threonine as a component of tRNA molecules [44,45], a decrease in its synthesis can disrupt both translation and protein assembly, thus bringing key processes of cellular metabolism to a complete halt.

Another group of proteins down-regulated in *A. hydrophila* after exposure to antimicrobial fish plasma are involved in transcriptional and post-transcriptional regulation, such as the Crp/Fnr family transcriptional regulator (A0KGX7), aconitate hydratase B (A0KPU4), 2-methylisocitrate dehydratase (A0KPU4), chaperone protein DnaJ (A0KMI5), two peptidyl-prolyl isomerases (A0KLW4, A0KHC2), ATP-dependent protease subunit HslV (A0KQI7), aminopeptidase P (A0KFS0), and the two-component regulatory system protein (A0KQC6) [46,47,48,49,50,51,52,53,54]. In addition to challenging the central regulation of metabolism in AP-treated bacteria, the inhibition of the synthesis of these proteins must also affect the adaptive mechanisms of the response to environmental stress by disrupting the energy production and maintaining the redox balance.

The composition of membrane-associated proteins was also altered in *A. hydrophila* treated with bacteriostatic AP in comparison to that in active-dividing bacteria. Acyltransferase family protein A0KNH8, adhesin-like protein A0KIU4, and the iron-siderophore ABC transporter substrate-binding protein (A0KJP4) were among the proteins unique to the Bac + AP proteome. In turn, iron, amino acid, and peptide ABC transporters (A0KG07, A0KR03, and A0KNG2, respectively), lipoproteins (A0KP92, A0KHC3), and outer membrane porin N (A0KLY9) associated with the plasma membrane were not identified in AP-treated *A. hydrophila*, but were present in Bac + MH and Bac + NP. Therefore, different components of the bacterial membrane transport system are notably affected after contact with AP, including monatomic ions, iron, amino acid, carbohydrate, and xenobiotic transport [55,56,57,58,59]. Changes in the cell wall and membrane proteins were expected because of the direct contact between the host and parasite molecules at the cell surface. However, there is insufficient data on the functions of these membrane proteins to conclude whether the change in their synthesis could have triggered the signalling cascade that led to the observed arrest in protein synthesis or the disruption of bacterial cell morphology and cell–cell adhesion.

## 4. Discussion

To better understand the natural variability of innate immunity in cultured fish, we evaluated the *in vitro* bactericidal activity of trout blood plasma against the fish pathogen *A. hydrophila*. We chose plasma fraction rather than serum because fibrinogen and coagulation factors may have antimicrobial activity [60,61] and be involved in the formation of antimicrobial complexes on the cell surface. However, it is important to note that the anticoagulants used for sample preparation are compounds that can interfere with the activity of plasma proteins, including those involved in the innate immune response and bacterial survival [62,63].

The bacteriostatic properties of fish plasma are known to vary significantly depending on species, physiological state, and environmental conditions [64]. It has previously been shown that plasma from non-immunised fish may [1,65] or may not [66] have antimicrobial activity. In cultured rainbow trout, the baseline serum-mediated bactericidal activity was reported to be pathogen-dependent: high levels have been reported against *A. hydrophila*, *Yersinia ruckeri*, *F. psychrophilum*, and *E. coli* and low levels against *A. salmonicida* and *Serratia marcescens* [64,67], although the latter could be increased via antigenic stimulation. However, our experiment showed that NP plasma from non-immunised trout did not inhibit the growth of *A. hydrophila*, demonstrating that antigen loading is necessary for the final deployment of the humoral immune response in cultured rainbow trout.

Chronic gastrointestinal disease, probably related to *F. psychrophilum* infection, is characteristic of trout whose plasma was shown to effectively suppress bacterial growth (AP). It is likely that the infection triggered the production of humoral immune factors and/or antibodies in AP, which had bacteriostatic effects. Similar relatively low specificity and agglutinative activity have been demonstrated previously for certain classes of immunoglobulins, such as IgM [68,69]. Therefore, circulating immunoglobulins, which were abundant in the blood of trout infected with *F. psychrophilum*, were suggested as likely candidates for the major antimicrobial factor in the plasma of immunised trout. A conserved region of certain bacterial motility and adhesion factors in *A. hydrophila* has been suggested as the target of the observed antibody cross-reactivity. 

In our experiment, a detailed study of the AP proteome showed that immunoglobulin isoforms from immunised trout differ in their ability to bind bacterial cells. The immunoglobulin heavy chain variable (IGHV) isoform A0A060Y462, which was abundant in AP plasma, was found to adhere to *A. hydrophila* cells treated with both high and low antimicrobial plasma. In turn, another IGHV isoform (A0A060YVB8), together with the immunoglobulin kappa light chain (A0A060Z062), was identified exclusively in the lysate of bacteria treated with bacteriostatic AP plasma. These proteins were not detectable in the plasma mass spectra, suggesting that minor but highly selective components of AP play a leading role in pathogen recognition.

In addition to the immunoglobulins, it was expected that complement components and acute phase proteins would be present in abundance in the fish plasma with high antimicrobial activity [1]. Indeed, the increased levels of complement components C4 and C1q (precerebellin) observed in the AP plasma were consistent with the inflammatory state of fish in the presence of foreign invaders [15,38,70]. However, the levels of other key membrane attack complex (MAC) components C1r, C3, C5, and C7, as well as certain PAMP recognition proteins (C-reactive protein, intelectin, etc.) were found to be lower in AP or at levels found in the plasma with low bacteriostatic properties. Trout from farm A have a long-lasting and moderately lethal gastrointestinal disease (as reported by the farmers); thus, the observed proteomic profile of AP may indicate a chronic phase of this infection, with an attenuated reaction of the acute phase of inflammation.

Analysis of fish proteins retained by *A. hydrophila* showed that both AP and NP contained major components of bactericidal MAC. Due to the presence of specific immunoglobulins, antibody-mediated MAC assembly was expected to be more effective in AP plasma than in NP. Furthermore, the properdin-like protein (A0A060XDG4), which promotes MAC assembly in the absence of an antibody, was only identified in lysates from bacteria treated with AP. However, the low lysis of *A. hydrophila* in the experiment suggests that the assembly of the membrane attack complex is unlikely to be the primary cause of the bacteriostatic effect of AP plasma.

Plasma lectins, which bind a wide range of bacteria and bacterial polysaccharides in fish, are other important plasma proteins with bacterial agglutination and antimicrobial activity. In AP plasma with a high bacteriostatic effect, we found increased levels of the C-type lectin (A0A060W1N4), a homologue of the fish-specific antimicrobial protein nattectin [71,72]. However, since this isoform was not detected in the washed lysate of either AP- or NP-treated bacteria, the direct antimicrobial activity of this protein was not confirmed *in vitro*. In turn, another PAMP-recognising C-type lectin, the ladderlectin-like protein (A0A060WRB3), was found to be retained in the lysate of *A. hydrophila* treated with antimicrobial AP plasma [73,74]. Notably, this protein was not identified in the plasma proteome, probably because of its low concentration and selective adherence to the bacterial surface. Fish ladderlectins have previously been reported to agglutinate and inhibit the growth of various Gram-positive and Gram-negative bacteria [75,76]; thus, it can be suggested as another candidate for the role of the antimicrobial factor in trout challenged with an *F. psychrophilum* infection.

According to the screening study of *A. hydrophila proteom*, treatment with trout plasma affects a number of aspects of bacterial metabolism, of which we believe that protein biosynthesis is the main target of fish antimicrobial factors. Although the bacteria were able to maintain cell integrity after AP treatment, a loss of components of the translation machinery was observed, indicating severe disorganisation of core cellular processes. This, in turn, could lead to further metabolic perturbations, and ultimately to the observed complete inhibition of cell growth and the disruption of *A. hydrophila* cell morphology. One possible mechanism for the observed cessation of protein production in *A. hydrophila* is the inhibition of threonine synthesis [45,77], as suggested by the absence of homoserine dehydrogenase and threonine synthase enzymes in AP-treated bacteria. However, it is not yet clear which host molecules might be the trigger for this process. Another non-lytic antimicrobial mechanism has previously been reported for proline-rich antimicrobial peptides (PrAMPs), which penetrate bacterial membranes and bind to the ribosome, completely blocking protein translation [78,79]. In our trout plasma protein database, we found no difference in the abundance of common precursors of PrAMPs, such as piscidins, cathelicidin, defensin, etc. [80,81], between the proteomes of high and low bacteriostatic plasma. Therefore, it is of further interest to investigate the rainbow trout plasma proteome to identify antimicrobial peptide precursor sequences that exhibit antimicrobial activity [82,83,84,85].

A proteomic study by Dong et al. [1], using a similar experimental scheme, showed a similar decrease in ribosomal proteins in *E. tagda,* treated with turbot *S. maximus* serum, together with the rapid down-regulation of proteins involved in energy metabolism, outer membrane assembly, and protein transport. A rapid decrease in proteins involved in antioxidant defence, such as catalase, has also been reported, suggesting that fish plasma may interfere with antioxidant defence and thus cause oxidative stress. Reactive oxygen species are known to damage ribosomal RNA and translation factors, disrupting the proper functioning of the ribosome [86,87,88]. However, we could not prove the oxidative effect of *O. mykiss* plasma on *A. hydrophila*, as we found no difference in antioxidant enzymes detected in the lysates of bacteria incubated in AP or NP plasma or the culture medium. Notably, thermostable factors were found in turbot serum to suppress *E. tarda* proliferation, whereas native serum caused membrane perforation, which is consistent with the abundance of many fish complement components and immunoglobulins retained on the bacterial surface. Thus, turbot and trout appear to have different humoral defence mechanisms, suggesting that different fish species could potentially be sources of different immunoactive compounds.

## 5. Conclusions

In our screening study, we found that antimicrobial proteins in the plasma of cultured rainbow trout infected with *F. psychrophilum* inhibited the growth of *A. hydrophila* by stopping protein biosynthesis in the bacteria rather than by causing cell perforation via the assembly of MACs. Cross-reactive antibodies and the antimicrobial protein ladderlectin, which is capable of cell agglutination, are thought to be responsible for the bacteriostatic properties of the fish plasma. Screening analysis of the *A. hydrophila* proteome when exposed to trout plasma with high antimicrobial activity revealed no specific defence mechanisms activated by the pathogen, which is probably due to the critical detrimental effect of fish immune molecules on bacterial metabolism.

During the study, we did not find a correlation between protein abundance in trout plasma and the composition of fish proteins that bind to *A. hydrophila*, but it is important to note that mass spectrometry bias may have limited our analysis of minor components of the plasma proteome. However, the affinity approach greatly expanded the range of plasma proteins analysed, which allowed the capture of minor components that selectively bind to antigens and are thus directly involved in immune defence. Therefore, studying the composition of protein complexes at the pathogen–host interface has been demonstrated to be informative for analysing antimicrobial factors in complex biological mixtures.

For future studies, we suggest that it would be useful to evaluate plasma composition after the removal of major proteins from plasma, which can mask the mass spectra of minor protein components, and to investigate the effect of anticoagulants on protein adhesion to the cell wall and MAC formation. Particular attention could also be paid to studying the composition of the bacterial membrane and the cell wall protein fraction as a zone of host–parasite interaction. Finally, in view of the data obtained, it would be interesting to study mutations in bacteria that become resistant to antimicrobial plasma. This would provide a better understanding of the mechanism of antimicrobial activity of plasma proteins.

## Figures and Tables

**Figure 1 animals-13-03565-f001:**
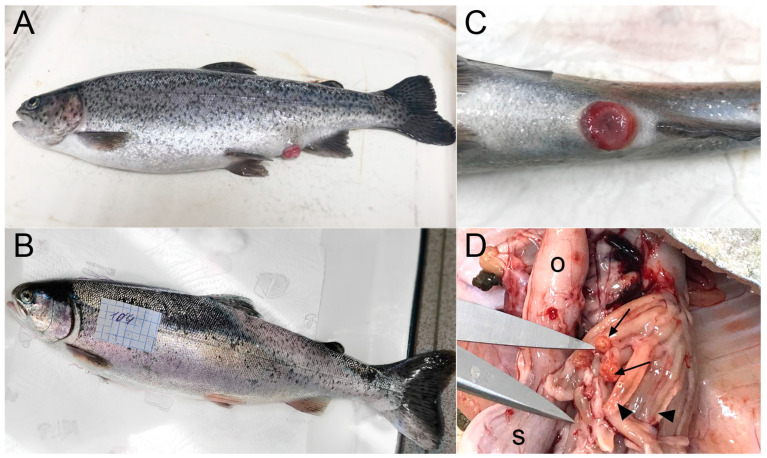
Rainbow trout *O. mykiss* collected for study. (**A**)—Trout with high antimicrobial activity in blood plasma from farm A exhibited enteritis and anal prolapse. (**B**)—Trout with low antimicrobial activity of blood plasma from farm N exhibited normal anatomy and morphology. (**C**)—Enlarged photograph of a swollen anus noted as a typical feature of trout disease in farm A. (**D**)—Transverse ulcerations (arrows) of the pyloric appendages (arrowheads) have been reported in fish from farm A, indicating severe damage caused by bacterial infection. Letter designations: s—stomach, o—oesophagus.

**Figure 2 animals-13-03565-f002:**
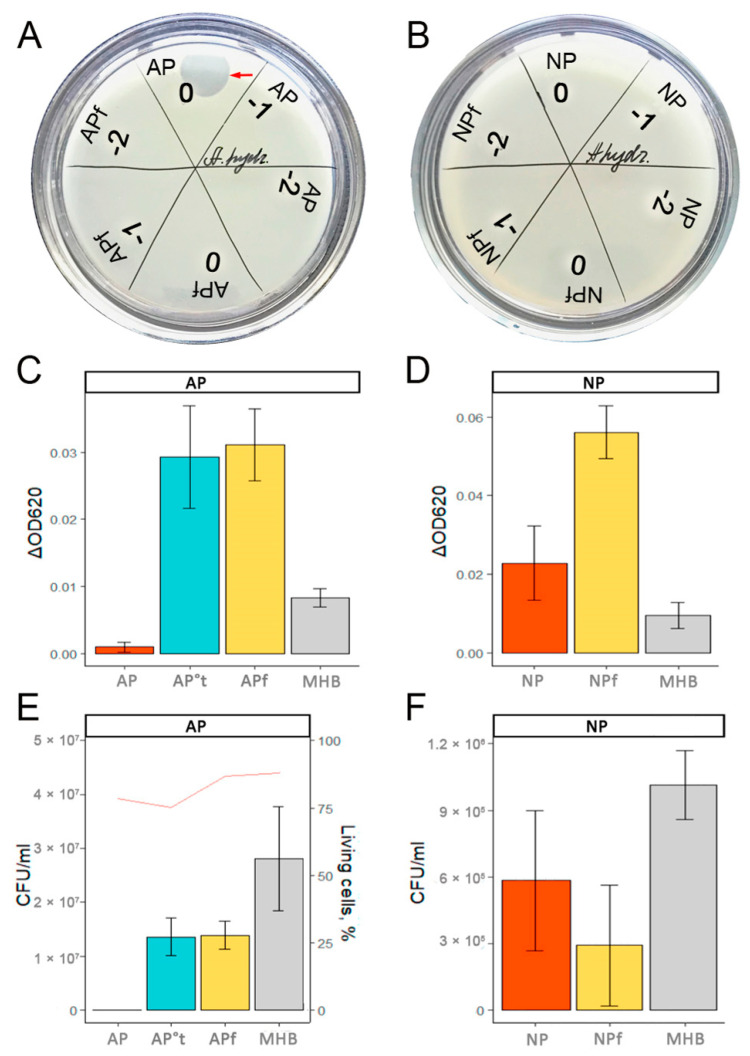
Effect of farmed trout plasma on the growth of *A. hydrophila* cultures. (**A**)—Inoculation of *A. hydrophila* on a petri dish with 30 µL of intact (AP) or filtered (APf, <10 kDa) trout blood plasma from farm A at different dilutions (0—undiluted, −1—10-fold diluted, and −2—100-fold diluted). (**B**)—Inoculation of *A. hydrophila* on a petri dish with 30 µL of intact (NP) or filtered (NPf, <10 kDa) trout blood plasma from farm N at different dilutions (0—undiluted, −1—10-fold diluted, and −2—100-fold diluted). (**C**)—Changes in optical density (S ± SE) of bacterial cultures after addition of intact (AP), heat-inactivated (AP°t), or filtered (APf, <10 kDa) fish blood plasma from farm A or Mueller–Hinton broth (MHB). (**D**)—Changes in optical density (S ± SE) of bacterial cultures after addition of intact (NP) or filtered (NPf, <10 kDa) fish blood plasma from farm N or Mueller–Hinton broth (MHB). (**E**)—Number of dividing colonies (S ± SE) transferred to solid culture media after incubation of bacteria for 4 h in intact (AP), heat-inactivated (AP°t), or filtered (APf, <10 kDa) fish plasma from farm A or in Mueller–Hinton broth (MHB). The red line shows the percentage of intact bacterial cells, as assessed using the LIVE/DEAD^®^ BacLight Bacterial Viability Kits. (**F**)—Number of dividing colonies (S ± SE) transferred to solid culture media after incubation of bacteria for 4 h in intact (NP) or filtered (NPf, <10 kDa) fish plasma from farm N or Mueller–Hinton broth (MHB).

**Figure 3 animals-13-03565-f003:**
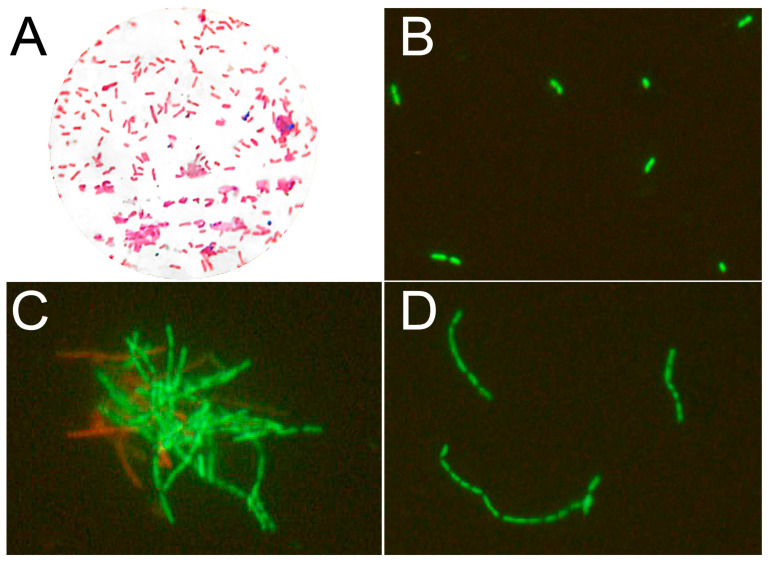
Morphology and cell integrity of *A. hydrophila* after 4 h treatment with trout plasma with high antimicrobial activity (AP). (**A**)—Morphology of bacteria in the culture medium, Gram staining. (**B**)—Bacteria that were incubated in the culture medium appeared as short bacilli that separated after division (BacLight Live/Dead cells stain). (**C**)—Agglutination of bacteria incubated in the native AP plasma. Green dye stains live cells with intact membranes and red dye stains dead cells with damaged membranes. (**D**)—Chains of bacterial cells adhering to each other when incubated in AP.

**Figure 4 animals-13-03565-f004:**
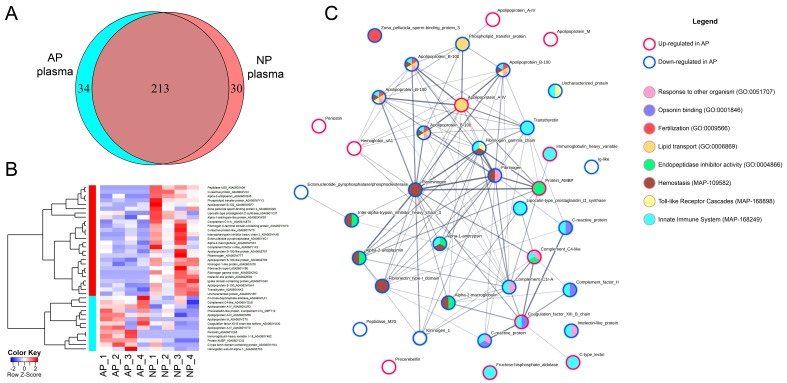
Comparison of the protein composition of blood plasma from *O. mykiss* with high (AP) and low (NP) bacteriostatic activity. (**A**)—Venn diagram of the qualitative protein composition of the plasma of *O. mykiss* shows the overlapping and unique proteins that were identified in the AP and NP. (**B**)—Heat map of *O. mykiss* proteins differentially expressed between the AP and NP plasma. (**C**)—Protein–protein interaction network for *O. mykiss* proteins differentially expressed in AP and NP, with some representative enriched GO ontologies and Reactome DB pathways.

**Figure 5 animals-13-03565-f005:**
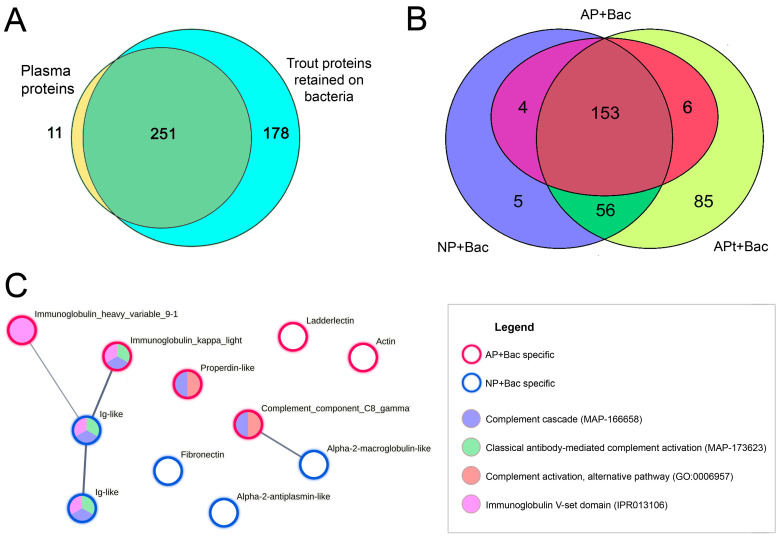
Profile of *O. mykiss* plasma proteins retained on *A. hydrophila* cells. (**A**)—Venn diagram of the qualitative protein composition of *O. mykiss* plasma and trout proteins retained on *A. hydrophila*. (**B**)—Venn diagram of the qualitative composition of trout proteins identified in washed suspensions of bacteria incubated in high (AP + Bac) and low (NP + Bac) bacteriostatic native or denatured (AP°t + Bac) trout plasma. (**C**)—Protein–protein interaction network for *O. mykiss* proteins identified in a washed bacterial suspension that was incubated in only one of two trout plasmas with high (AP + Bac) or low (NP + Bac) bacteriostatic activity.

**Figure 6 animals-13-03565-f006:**
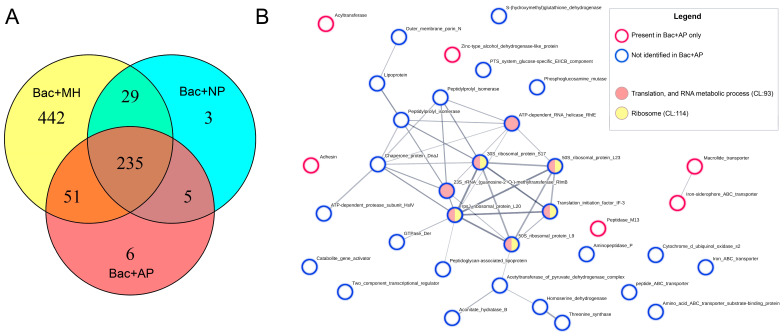
Protein profile of *A. hydrophila* after treatment with trout plasma. (**A**)—Venn diagram of the *A. hydrophila* protein composition after 4 h of incubation in MHB culture medium (Bac+MH), trout plasma with high (Bac + AP) or low (Bac + NP) bacteriostatic activity. (**B**)—Protein–protein interaction network for *A. hydrophila* proteins identified only in bacteria incubated in trout plasma with a high bacteriostatic effect or not identified in Bac + AP, although it was found in bacteria treated with Bac + NP and Bac + MH.

## Data Availability

Data are contained within the article and Appendix A. All mass spectrometry data are deposited in ProteomeXchange with the identifier PXD037789, PXD043655, and PXD044468. The sequence raw data and resulting OTUs are available on the Mendeley platform at ncbi.nlm.nih.gov/bioproject/PRJNA897820 (accessed on 10 November 2023).

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
