# Peer review of "Antibacterial Activity of Rainbow Trout Plasma: In Vitro Assays and Proteomic Analysis"

_animals, 2023, doi:10.3390/ani13223565_

Round 1

Reviewer 1 Report

Comments and Suggestions for Authors

The authors Mirzaeva et al investigated the Rainbow trout plasma for its antibacterial activity. I would suggest the authors to consider some points discussed below and revise the manuscript accordingly.

The authors have to check the spellings of some words including the documentary files. Spellings need to be corrected before publishing.

Expand the LFQ in keywords

The scientific names in the manuscript need to be italicized. Some of the scientific names are not in italics. Authors have to check the whole manuscript.

Please include the product length of the primers used to amplify hypervariable v3-v4 region.

For the figure 2, check the petriplate dilution, there is a mention two times -1 in A and B panels. Why the authors have not identified the living cells for NP in Panel F?

Author Response

Dear Reviewer 1,

Thank you very much for your time and effort during review of our manuscript “Antibacterial activity of rainbow trout plasma: in vitro assays and proteomic analysis”. We tried our best to fulfill the suggested corrections.

Please find our point-by-point comments below.

Reviewer 1: The authors Mirzaeva et al investigated the Rainbow trout plasma for its antibacterial activity. I would suggest the authors consider some points discussed below and revise the manuscript accordingly.

“The authors have to check the spellings of some words including the documentary files. Spellings need to be corrected before publishing.”

We appreciate your careful review of the spelling of certain terms within our manuscript. We have addressed the errors in the text, including those found in the Supplementary Materials section (lines 28, 104, 115, 118, 120, 128, 139, 141, 256-257, 259-262, 330, 338, 421, 441,  506, 615,  630,633,  635, 670, 740, 772-781). 

R1: “Expand the LFQ in keywords”

We are grateful for your meticulous focus on the abbreviations. After reviewing your feedback, we have revised our manuscript to expand "LFQ" to "label-free quantitation."

R1: “The scientific names in the manuscript need to be italicized. Some of the scientific names are not in italics. Authors have to check the whole manuscript.”

We appreciate your careful consideration of the scientific names in our manuscript. We have made the necessary changes to the manuscript. 

R1: “Please include the product length of the primers used to amplify hypervariable v3-v4 region”

Information on the length of the product (428 bp) has been added to the Materials and Methods.

R1: “For the figure 2, check the petriplate dilution, there is a mention two times -1 in A and B panels. 

Thank you for your attention, we have double checked and found that Figure A2 shows incorrect actual dilutions. The labels on the figure have been corrected.

R1:  Why the authors have not identified the living cells for NP in Panel F?

Thank you for your comment! The experimental design used was to test whether native AP plasma has bacteriolytic activity; accordingly, bacteria whose growth was inhibited by AP plasma with antimicrobial activity were selected as the experimental group and bacteria treated with denatured or filtered plasma or culture medium without antimicrobial activity were used as the control groups. Investigating the effect of NP plasma as an additional control group would certainly provide more information on the variability of cell survival parameters exposed to native plasma, but would also make the results more difficult to interpret due to possible differences in the chemical composition of AP and NP.

We agree that it would be interesting to check the number of living cells after NP treatment to investigate the overall effect of native trout plasma on bacteria survival, but this was not the purpose of this particular experiment.

 Best regards,

Mr. Stanislav Rimaso Kurpe

Reviewer 2 Report

Comments and Suggestions for Authors

This study aimed to investigate the ability of the proteins involved in the natural antibacterial activity of blood plasma from rainbow trout obtained from two different fish farms. The results demonstrated that the plasma from naturally infected trout effectively suppressed the growth of Aeromonas hydrophila while preserving the integrity of the cell membrane. The analysis identified specific proteins within the plasma that exhibited notable antimicrobial properties, including apolipoproteins, immunoglobulins, complement components, coagulation factors, lectins, periostin, and hemoglobin. Furthermore, the investigation delved into the bacterial response to trout plasma and suggested that the target of these antimicrobial proteins derived from fish blood was the protein synthesis pathway. These findings underscore the vital role of plasma proteins in the host's defense against pathogens, shedding light on their significance in natural immunity.

The title clearly reflects the contents of the paper

The abstract is informative

The introduction is ok and properly reviewed the relevant literature. Also, the objectives clearly stated.

The methods are not clear, and the below concerns should be addressed.

The results presented are adequate.

The tables and figures used to show them are adequate.

The discussion is fairly comprehensive.

The conclusion effectively aligns with the study's results and clearly articulates the primary hypothesis. However, it is advisable for the authors to consider making recommendations for future research endeavors.

 Therefore, this study, which delves into the immune response exhibited by the blood plasma of trout, offers valuable insights. Nonetheless, I would like to encourage the authors to address the following concerns:

 Ensure that all scientific names in the text are appropriately italicized.

Clarify the rationale behind examining only four fish per cage. The sample size appears relatively small, and I wonder if it was adequate.

A notable absence in the manuscript is the lack of statistical analysis, which is typically a fundamental component of scientific studies.

It would be valuable for the authors to provide an explanation for their choice of plasma over serum. It's important to note that the use of anticoagulants can potentially impact the functions of proteins and enzymes, thereby potentially affecting bactericidal activity.

Correct an inaccuracy in line 128: Clove oil is emulsified in water, and the term "suspension" is not accurate in this context.

Author Response

Dear Reviewer 2,

Thank you very much for your time and effort during review of our manuscript “Antibacterial activity of rainbow trout plasma: in vitro assays and proteomic analysis”. We tried our best to fulfil the suggested corrections.

Please find our point-by-point comments below.

Reviewer 2: “This study aimed to investigate the ability of the proteins involved in the natural antibacterial activity of blood plasma from rainbow trout obtained from two different fish farms. The results demonstrated that the plasma from naturally infected trout effectively suppressed the growth of Aeromonas hydrophila while preserving the integrity of the cell membrane. The analysis identified specific proteins within the plasma that exhibited notable antimicrobial properties, including apolipoproteins, immunoglobulins, complement components, coagulation factors, lectins, periostin, and hemoglobin. Furthermore, the investigation delved into the bacterial response to trout plasma and suggested that the target of these antimicrobial proteins derived from fish blood was the protein synthesis pathway. These findings underscore the vital role of plasma proteins in the host's defense against pathogens, shedding light on their significance in natural immunity.

The title clearly reflects the contents of the paper

The abstract is informative

The introduction is ok and properly reviewed the relevant literature. Also, the objectives clearly stated.

The methods are not clear, and the below concerns should be addressed.

The results presented are adequate.

The tables and figures used to show them are adequate.

The discussion is fairly comprehensive.

The conclusion effectively aligns with the study's results and clearly articulates the primary hypothesis. However, it is advisable for the authors to consider making recommendations for future research endeavors.”

We appreciate your suggestions for improving the text. We have added recommendations for future research at the end of the conclusion.

R2:  “Therefore, this study, which delves into the immune response exhibited by the blood plasma of trout, offers valuable insights. Nonetheless, I would like to encourage the authors to address the following concerns:

Ensure that all scientific names in the text are appropriately italicized.”

We are grateful for your attention to the scientific names in our manuscript. We have made the necessary changes to the manuscript. 

R2: “Clarify the rationale behind examining only four fish per cage. The sample size appears relatively small, and I wonder if it was adequate.”

Thank you for your valuable comment! You raise valid concerns about adherence to scientific methodology. In short, a sample size of 4 is small for parametric statistics, but may be sufficient when using statistical approaches to process omics data. Therefore, in the absence of clear and reasonable recommendations for omics studies today, one should be guided by common sense, cost-effectiveness and ethics.

In fact, there is no specific minimum statistical sample size for omics studies, and a sample size of n=3-4 is common in biological studies (Weththasinghe et al., 2021; Reveco-Urzua et al., 2019; Ortiz-Severín et al., 2020; Shi et al., 2019). Due to their typically small sample size and the very specific nature of omics data (covariate gene expression and a much larger number of evaluated parameters (thousands) than the number of samples), specific statistical approaches and integrated solutions have been (and are being) developed (Ritchie et al., 2015). These sets of principles improve statistical power and accuracy by better modelling the characteristics of the data, ensuring that inference is reliable and stable even when the number of replicates is small. We used the Limma statistical protocol for differential gene/protein expression analysis, which, among other things, takes into account the strength of interactions between genes and also estimates the trend of the mean variance for data that are less reliable at low concentrations (typical of mass spectrometry)(Ritchie et al., 2015). Limma uses a class of statistical methods called empirical Bayes to estimate the prior probability distribution from the data, and then uses linear models (moderated t-statistics) for group comparisons, resulting in increased statistical power to detect differential expression.

Within this minimum sample problem, a very important question is whether we want to draw conclusions about specific samples or about the whole population. In clinical trials, the choice of sample size selection is critical for identifying biomarkers that are valid in a heterogeneous population (Zhang et al., 2019, doi:10.1088/1755-1315/252/2/022127). The minimum observed number of samples that may provide sufficient statistical power to find biomarkers in human plasma proteins was estimated to be six (8-plex iTRAQ-labelled proteomic, log2 fold change > 1) based on an a priori power analysis of protein variability in the population.

 While increasing the sample size is often a priority in biomarker discovery experiments to enhance the chances of identifying valid biomarkers, it's important to note that our study had a different focus. In our study we were looking for differences between groups of samples and not to study protein variation in trout populations (the two populations studied were genetically homogeneous with fish sharing a very common background). Nevertheless, in our study, differentially expressed proteins were filtered using p-values <0.05 adjusted for multiple testing using the Benjamini-Hochberg method and log2 fold change > 1 to reduce the risk of false positives. To make it clearer, we have now tried to reflect the screening nature of the study in the text of the article.

Pilot studies such as ours are essential for estimating variability and effect size within a particular experimental system. This information is valuable for sample size calculations in subsequent studies. To obtain more global biological results, for example for biomarker identification, a second round of testing after a screening study may be recommended (Kirpich et al., 2018, https://doi.org/10.1371/journal.pone.0197910).

In conclusion, we agree that a larger sample size is preferable to increase the statistical power of the analysis and reduce the risk of false positive or false negative results. In our pilot study, we aimed to lay the groundwork for future investigations of protein-protein interactions between pathogens and hosts, so we chose 4 fish as the minimum sample size, which is cost effective and sufficient to complete the tasks set.

R2: “A notable absence in the manuscript is the lack of statistical analysis, which is typically a fundamental component of scientific studies.”

Complete statistical analysis based on t-statistic, which is common in gene expression comparison, was used to analyse the differential expression of proteins in AP and NP plasma as described above.

The choice of qualitative comparison of some samples was made due to the technical limitations of mass spectrometry in terms of protein concentration and to reduce the number of false positives. We were unable to collect sufficient protein in each replicate in vitro assay for the analysis of bacterial proteins and anchored trout proteins. Therefore, samples from three biological replicates of the experiment were pooled into one, so that we believe the variability caused by experimental conditions was accounted for. Each sample was analysed in technical triplicates to eliminate data variability due to the chromatography-mass spectrometry procedure. However, as statistical analyses are sensitive to spurious increases in statistical power due to sample duplication, we have chosen not to report the results of differential expression analyses based on technical replicates. The article only discusses qualitative differences between these samples, which in a static sense are identical to the results of differential expression analysis with the strictest cut-off of results at the maximum log2 fold change value.

For clarity, we have added a more detailed description of the analysis steps in the Materials and methods section.

R2: “It would be valuable for the authors to provide an explanation for their choice of plasma over serum. It's important to note that the use of anticoagulants can potentially impact the functions of proteins and enzymes, thereby potentially affecting bactericidal activity.”

We chose plasma because we considered that fibrinogen and coagulation factors may also be involved in antimicrobial activity and the formation of antimicrobial complexes on the cell surface (Påhlman et al., 2013; Prasad et al., 2021). Relevant comments have been added to the text in the first paragraph of the Discussion. We were also concerned that a significant proportion of the proteins would be lost as they are entrained in the sedimentation of the fibrin clot. 

It's important to note that the use of anticoagulants can potentially impact the functions of proteins and enzymes, thereby potentially affecting bactericidal activity.

Thank you for this feedback. We have noted the potential effect of anticoagulants on bactericidal activity in the first paragraph of the Discussion.

R2: “Correct an inaccuracy in line 128: Clove oil is emulsified in water, and the term "suspension" is not accurate in this context.”

Thank you for your attention to this matter. We acknowledge the inaccuracy in our manuscript where we referred to "suspension" instead of the correct term "emulsification" regarding the clove oil in water. The sentence “On the day of sampling, the fish were anaesthetized with an aqueous emulsified clove oil in water” was replaced to “On the day of sampling, the fish were anaesthetized with clove oil emulsified in water” (line 128-129)

Best regards,

Mr. Stanislav Rimaso Kurpe

Round 2

Reviewer 2 Report

Comments and Suggestions for Authors

-